# Spatio-Temporal Evolution and Future Simulation of Agricultural Land Use in Xiangxi, Central China

**Hui Xiang** [1,2,3,4], **Yinhua Ma** [1,2,3,4], **Rongrong Zhang** [1,2,3,4], **Hongji Chen** [1,2,3,4] and **Qingyuan Yang** [1,2,3,4,*]

1   School of Geographical Sciences, Southwest University, Chongqing 400715, China; xh982123@email.swu.edu.cn (H.X.); yhm82@email.swu.edu.cn (Y.M.); zr1473135947@email.swu.edu.cn (R.Z.); chj001292@email.swu.edu.cn (H.C.)
2   Chongqing Jinfo Mountain Field Scientific Observation and Research Station for Kaster Ecosystem, Ministry of Education, Chongqing 400715, China
3   State Cultivation Base of Eco-Agriculture for Southwest Mountainous Land, Southwest University, Chongqing 400715, China
4   Chongqing Key Laboratory of Karst and Environment, Chongqing 400715, China
*   Correspondence: yizyang@swu.edu.cn; Tel.: +86-135-0949-6095

**Abstract:** Researches on agricultural land use would help the stakeholders to make better decisions about agricultural resources. However, studies on agricultural land have been lacking. In this context, Xiangxi was chosen as a typical region, and five indicators (Kernel Density, change importance, etc.) and two models (gray forecasting model and GeoSoS-FLUS) were used, to explore the spatio-temporal evolution trends and simulate the future scenarios of agricultural land use. The results were as follows: (1) Xiangxi was dominated by agricultural land, and nearly 50% of total extent was forestry land. Extent of agricultural land decreased by about 56.89 km$^2$ or 3.74% from 2000 to 2018; (2) The density of each agricultural land in the study area had considerable spatial heterogeneity, and showed a main trend of shrinkage, especially in the south regions; (3) In 2030, the spatial pattern and composition of agricultural land in Xiangxi will maintain the existing status, while both of the area and proportion of agricultural land will decline, with a loss of 241.34 km$^2$ or 2.85% decrease from 2000. Nevertheless, the study believed that the slight shrinkage of the agricultural land in Xiangxi is in line with the objective law. At the same time, the study suggested to strengthen the scientific management and rational utilization of agricultural land, with emphasis on arable land and fishery land in the south, especially the administrative center.

**Keywords:** agricultural land; land use change; spatio-temporal evolution; future simulation; Xiangxi

## 1. Introduction

Agriculture can refer to general agriculture (i.e., planting, animal husbandry, forestry, fishery, and others) and special agriculture (i.e., planting) [1]. This study uses the first definition. Agricultural land is land used by humans to intervene in the growth and development of organisms in the natural ecosystem to produce agricultural products [2]. It has both natural and economic characteristics. It is the place of agricultural practices [3], the main carrier of economic-social activities for rural population [4], and also the basic guarantee for material production and civilization development [5,6]. According to the differences between land types and products, it can be divided into arable land, forestry land, rangeland, and fishery land.

Land use change has attracted increasing attention from scientists around the world. Assessment technologies of land use change are RS and GIS, which have been applied to Malaysia [7], Qom in Iran [8], the middle Hills of Nepal [9], Italy [10], and promoted the effective utilization and rational allocation of land resources; the main simulation models of land use include CA-Markov [11,12], FLUS [13,14], etc. FLUS model, which can realize simulation and comparison according to the spatio-temporal dynamics of land use, is

the most popular and widely used model. However, the studies on land use change are carried out from land [15–17] or construction land [18–20], while studies on agricultural land have been lacking. Land use change is the overall reflection of human activities, which is conducive to the overall planning and management of land. Agricultural land is the extension and expansion of land, but the overall macro investigations of land cannot guide the local fine layout of agricultural land.

Agricultural land use change can not only alter the biophysical surface characteristics, but also lead to changes of climate, ecosystem services, and biological diversity [6,7]. As a result, researches on agricultural land use would help land managers, scientists, and other stakeholders to make better decisions about agricultural resources, which is not only conducive to improve the agricultural production capacity, ensure human health and reproduction [8,9], but also maintain the health and sustainability of the agricultural ecosystem [10,11]. As a result, there is a need for a comprehensive quantification and systematic assessment for agricultural land.

A small number of studies were conducted from the perspective of arable land and rangeland, while scientific literature on other types of agricultural land has been scarce. Alijani et al. [21] conducted a field research in Mazandaran province of Iran where they found that the most important factors in reducing the area of arable land were difficult economic conditions and the growth of the tourist industry. The empirical case integrated census and satellite data in Brazilian Amazonia [22] and indicated an overall expansion of agricultural area between 1980 and 1995. Among them, arable land and artificial rangeland increased, while natural rangeland decreased. With the changes in natural environment and social economy, the area and proportion of different agricultural land show different dynamic trends. Nevertheless, few studies have used spatial analysis technology and FLUS model to explore dynamic characteristics and future scenarios of different agricultural land. Furthermore, forestry land and fishery land, which are also important to humans, have received little attention.

In view of this, a characteristic agricultural production area, named Xiangxi, located in Hunan Province of central China, is chosen as a typical region to explore the spatio-temporal evolution trends and simulate the future scenarios of agricultural land. The specific objectives of this study are as follows:

1. The overall change pattern of agricultural land from 2000 to 2018 in Xiangxi is identified by spatial analysis methods.
2. The change characteristics of each agricultural land are detected using GIS techniques in the study area during the same study period.
3. The future scenarios of agricultural land use in 2030 are simulated by gray forecasting model and GeoSoS-FLUS model.

This research attempts to add the spatial information of agricultural land use change and provide decision-making reference for land managers. First, forestry land and fishery land are analyzed, and spatial analysis methods, gray forecasting model, and FLUS model are used. These are novel and of academic interest, which can provide reference for other scholars. Second, it may provide useful suggestions for the utilization and protection of agricultural resources, and contribute to the high-quality development of agriculture.

## 2. Data and Methods

### 2.1. Study Area

Xiangxi (full name: Xiangxi tujia and miao Autonomous Prefecture) [23] lies in central China, northwest of Hunan Province (Figure 1). It covers an area of 15,462 km$^2$ and is located at 109°10′ E–110°22′ E and 27°44′ N–29°38′ N. The region comprises eight administrative units and the administrative center is Jishou. The area has a subtropical monsoon climate, with a mean average annual temperature of about 16~18 °C and an average annual precipitation of about 1500 mm.

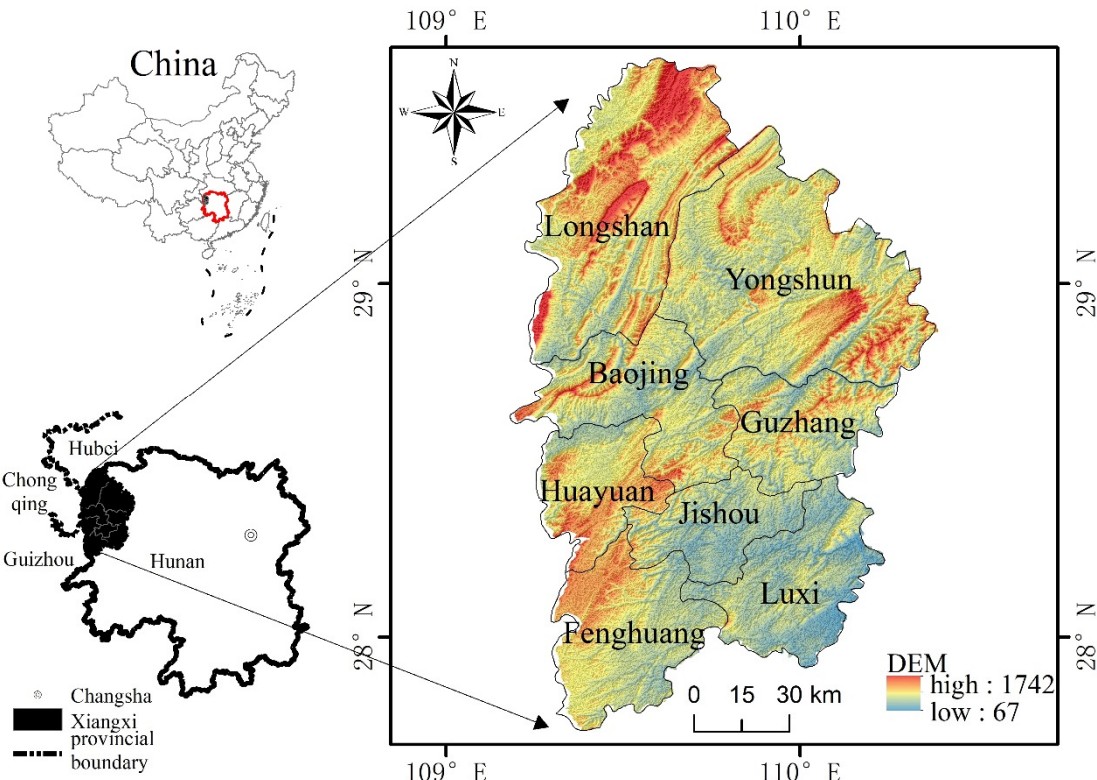

**Figure 1.** Location of the study area.

As a traditional area with a relatively backward economy, agriculture is the leading sector in Xiangxi; it is a typical mountainous region, and agricultural production has obvious particularity and heterogeneity. In 2014, the "Regional Layout Plan for Special Agricultural Products (2013–2020)" was issued, Xiangxi was planned as a special production area of traditional medicine and animal husbandry. In 2018, "Strategic Plan for the Revitalization of Rural Areas in Hunan Province (2018–2022)" was promulgated and Xiangxi was selected as the characteristic agricultural industrial belt. In this context, the agricultural characteristic industry has developed rapidly. At the same time, Xiangxi is in the key period of new rural construction and agricultural modernization. Nevertheless, the structure of land use in Xiangxi has undergone significant changes and construction land has expanded sharply. As a result, the problem of agricultural land protection has become very serious, and the contradiction between land supply and land demand is becoming increasingly prominent.

Therefore, the distribution pattern and dynamic evolution of agricultural land in Xiangxi are different from other regions. It represents a typical research case, which can provide scientific references for the optimal allocation of land use and the development of agricultural modernization in similar areas.

### 2.2. Datasets and Their Preprocessing

The datasets used are natural environmental data (DEM, temperature, precipitation, land use, rivers, etc.) and socio-economic data (administrative divisions, population density, railways, highways, urban centers, etc.). The 30 m resolution DEM data is obtained from the website of the United States Geology Survey (http://earthexplorer.usgs.gov/). (accessed on 5 April 2020).The slope and aspect characteristics are extracted through the spatial analysis tool of ArcMap10.2(It was invented by American Environment Systems Research Institute, and download from this website: http://www.esri.com), (accessed on 5 April 2020); the datasets of temperature, precipitation, rivers, railways, highways, and urban centers are downloaded through the software of Bigemap2021 (It was invented by Chengdu bigtu data processing corporation, and download from this website: http://www.bigemap.com), (accessed on 5 April 2020), and analyzed by the tool of European distance; the 30 m

resolution land use database are gathered from the Resources and Environmental Sciences Data Center of the Chinese Academy of Sciences (http://www.resdc.cn/), (accessed on 5 April 2020), and reclassified by ArcMap10.2; the administrative boundaries are collected from the National Earth System Science Data Sharing Platform (http://www.geodata.cn/), (accessed on 5 April 2020), and extracted according to the attributes; population data is taken from the statistical yearbook in Xiangxi.

Both vector data and grid data are used in our research. Firstly, vector data is converted into grid data through the conversion tool of ArcMap10.2. Secondly, the coordinate system of grid data is set as Krasovsky 1940 Albers. Finally, grid data is resampled to a uniform accuracy (30 m).

*2.3. Research Methods*

2.3.1. Agricultural Land Classification System Is Constructed

There are 6 first-class land types and 18 s-class land types in the study area. According to relevant research results [24–28], agricultural land is subdivided into four categories (arable land, forestry land, rangeland, and fishery land), and its classification system is constructed (Table 1). Based on the classification system, the land use data are reclassified.

**Table 1.** Agricultural land classification system.

| Class | Code and Name | Description | Basis for Classification |
|---|---|---|---|
| agricultural land | 1—arable land | paddy field; dry land | paddy field and dry land are the main farming sites and belong to arable land |
| | 2—forestry land | timber forest; economic forest; etc. | timber forest, economic forests, etc., are the main land for forestry production |
| | 3—rangeland | high, medium, low coverage grassland | grassland is divided into high, medium, and low coverage grassland, which is the main grazing place |
| | 4—fishery land | reservoirs and ponds | reservoirs and ponds are the main fishery place in mountainous areas |

2.3.2. Agricultural Land Use Change Is Assessed

Five indicators (Kernel Density, change importance, change area, change rate, and dynamic) are selected to reflect evolution trend of agricultural land in the study area. Among them, Kernel Density Estimation (KDE) cannot be affected by the resolution and location of data, which is used to achieve high-accuracy density measurement and has been applied in many fields [29–32]. The bandwidth is defined as the calculation range, and the spatial distribution of the elements is estimated based on the density of the observed objects in the area. The closer the element is to the center point, the greater the weight, and vice versa. The average density of all elements is the density of the observation points. The equation of KDE ($P_i$) is as follows:

$$P_i = \frac{1}{n\pi R^2} \times \sum_{j=1}^{n} K_j \left(1 - \frac{D_{ij}^2}{R^2}\right)^2 \tag{1}$$

where $K_j$ represents the weight of pixel $j$; $D_{ij}$ is the distance from pixel $i$ to pixel $j$; $R$ is the bandwidth ($D_{ij} < R$), which is 2 km in this study; $n$ is the number of pixel $j$ within the calculation range.

The index of change importance ($C_{impor}$) is chosen to detect the main change types or areas of agricultural land, and the equation is as follows:

$$C_{impor} = \frac{\Delta S}{\sum \Delta S} \times 100\% \tag{2}$$

where the value of $C_{impor}$ is between 0 and 1, the larger the value, the more dominant the change; $\Delta S$ is the change area of each agricultural land; $\Sigma\Delta S$ is the total change area of agricultural land.

The equation for calculating change rate of agricultural land ($N_a$) is as follows:

$$Na = \frac{S_b - S_a}{S_a} \times 100\% \tag{3}$$

where the larger the value of $N_a$, the faster change rate; $S_b$ and $S_a$ are the area of agricultural land at the final and the initial stage of the monitoring period, respectively.

The equation for calculating the dynamic of agricultural land ($K_d$) is:

$$K_d = \sum \left( \frac{|\Delta S|}{S_{sum}} \right) \times 100\% \tag{4}$$

where the larger the value of $K_d$, the more violent agricultural land change, and vice versa; $|\Delta S|$ is the absolute value of change area in agricultural land; $S_{sum}$ is the total area of land.

2.3.3. Future Scenario of Agricultural Land Use Is Simulated

Gray forecasting model and GeoSoS-FLUS model are used to simulate the future scenario of agricultural land use.

(1)    Gray forecasting model

Equidistant observations are used in the gray forecasting model to predict systems with uncertain factors [33,34]. The agricultural land dataset of 2000, 2005, 2010, and 2015 are used as basic data to predict the amount of agricultural land in 2030. The main process of establishing the single-sequence linear differential equation GM(1,1) is as follows:

First, a new data sequence ($x^{(1)}$) is obtained by accumulating the original data sequence ($x^{(0)}$).

$$x^{(0)} = \left( x^{(0)}(1), x^{(0)}(2), \ldots, x^{(0)}(n) \right) \tag{5}$$

$$x^{(1)}(t) = \sum_{k=1}^{t} x^{(0)}(k), t = 1, 2, \cdots, n \tag{6}$$

Second, the first order linear differential equation of new data sequence ($x^{(1)}$) is solved.

$$\frac{\mathrm{d}x^{(1)}}{\mathrm{d}t} + ax^{(1)} = v \tag{7}$$

Third, the mean value of the accumulated data $B$ and constant vector $Y_n$ are calculated.

$$B = \begin{bmatrix} 0.5\left(x^{(1)}(1) + x^{(1)}(2)\right) \\ 0.5\left(x^{(1)}(2) + x^{(1)}(3)\right) \\ \vdots \\ 0.5\left(x^{(1)}(n-1) + x^{(1)}(n)\right) \end{bmatrix} \tag{8}$$

$$Y_n = \left( x^{(0)}(2), x^{(0)}(3), \cdots, x^{(0)}(n) \right) \tag{9}$$

Fourth, the gray parameter $\hat{a}$ is solved.

$$\hat{a} = \begin{pmatrix} a \\ v \end{pmatrix} = \left( B^T B \right)^{-1} Y_n \tag{10}$$

Fifth, the prediction function is solved.

$$\hat{x}^{(1)}(t+1) = \left( x^{(0)}(1) - \frac{v}{a} \right) e^{-at} + \frac{v}{a} \tag{11}$$

Sixth, $\hat{x}^{(0)}$ is obtained by reducing $X^{(0)}$.

$$\hat{x}^{(0)}(t+1) = \hat{x}^{(1)}(t+1) - \hat{x}^{(1)}(t) \tag{12}$$

(2)    GeoSoS-FLUS model

The advantages of GeoSoS-FLUS are wide applicability and high resolution [13,35]. Artificial neural networks (ANN) and cellular automata (CA) are used in GeoSoS-FLUS model to simulate land use change scenario. The main calculation steps are as follows.

First, the driving factors of agricultural land use change are identified. Agricultural land can be influenced not only by natural variables such as topography and climate [36], but also by socio-economic variables such as demographic and traffic drivers [24]. According to the related reference [37,38], 10 important driving factors are selected from two aspects: natural and human variables (Table 2, Figure 2).

**Table 2.** Driving factors of agricultural land use change.

| Data Categories | Data Name | Year | Unit | Reference |
|---|---|---|---|---|
| natural factors | elevation | | m | [37] |
| | slope | | ° | [37,38] |
| | aspect | | ° | [37] |
| | temperature | 2018 | °C | [37] |
| | precipitation | 2018 | mm | [38] |
| human factors | population density | 2018 | person/km$^2$ | [37] |
| | distance to highways | 2018 | m | [37,38] |
| | distance to railways | 2018 | m | [37] |
| | distance to the cities | 2018 | m | [37,38] |
| | distance to rivers | 2018 | m | [37,38] |

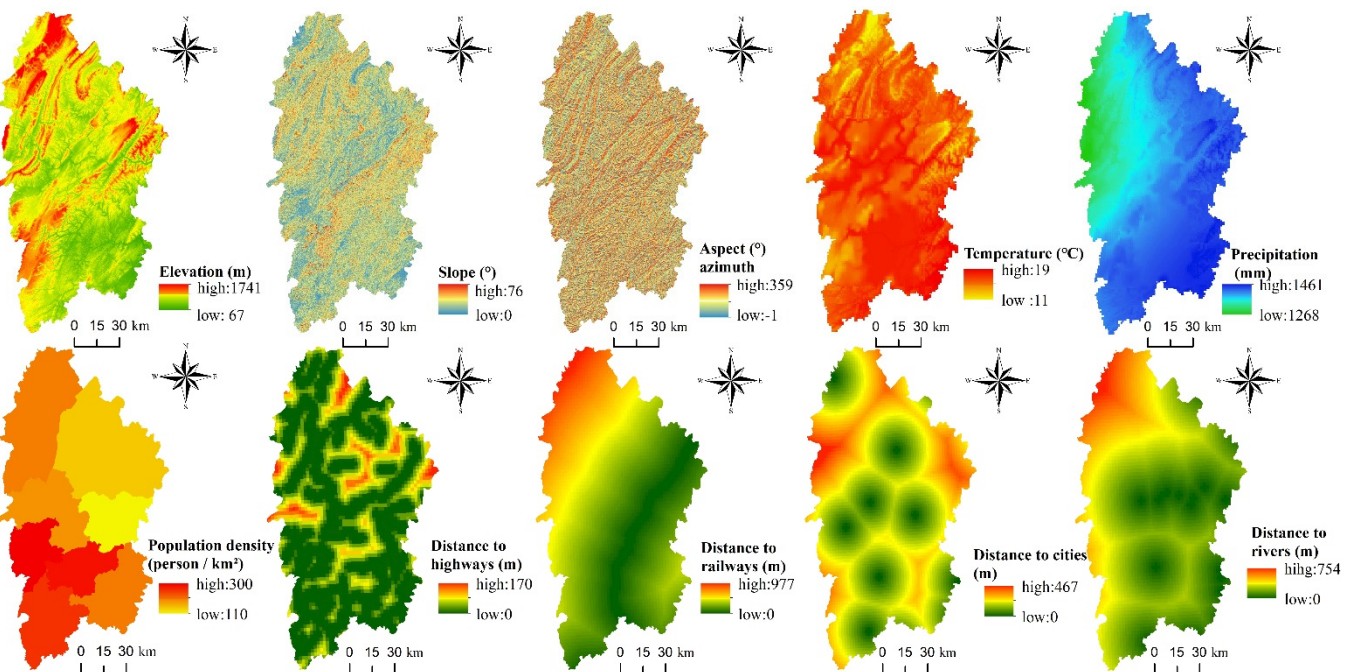

**Figure 2.** Driving factors of agricultural land use change.

Next, the suitability probability is calculated based on ANN. The complex relationship between input data and setting targets is continuously simulated through lots of learning and iteration by ANN, to calculate the suitability probability of land. The calculation equations are as follows:

$$sp(p,k,t) = \sum_j \omega_{j,k} \times sigmoid\left(net_j(p,t)\right)$$

$$= \sum_j \omega_{j,k} \times \frac{1}{1 + e^{-net_j(p,t)}} \tag{13}$$

$$\sum_k sq(p,k,t) = 1 \tag{14}$$

In the above equations, $sp(p,k,t)$ represents the suitability probability of land *K* in grid *P* at time *t*, and the sum of suitability probability for all land types is 1; $\omega_{j,k}$ represents the adaptive weight between the output and the hidden layers, which is adjusted according to the training results; *sigmoid* is the correlation function between the hidden and the output layers; $net_j(p,t)$ is the signal received by the hidden layer *j* grid *p* at time *t*.

Then, the adaptive inertia competition mechanism is evaluated based on CA model. The conversion probability is affected by the adaptive inertia coefficient of land. The equation of the adaptive inertia coefficient for land *K* at time *t* ($Intertia_k{}^t$) is as follows:

$$Intertia_k^t \begin{cases} Intertia\,_k^{t-1}, \left|D_k^{t-2}\right| \leqslant \left|D_k^{t-1}\right| \\ Intertia\,_k^{t-1} \times \frac{D_k^{t-2}}{D_k^{t-1}}, 0 > D_k^{t-2} > D_k^{t-1} \\ Intertia_k^{t-1} \times \frac{D_k^{t-1}}{D_k^{t-2}}, D_k^{t-1} > D_k^{t-2} > 0 \end{cases} \tag{15}$$

where $D_k^{t-2}$ and $D_k^{t-1}$ represent the demand and the actual quantity of land *k* at time $t-1$ and $t-2$, respectively.

The equations of the probability for grid *P* converting into land *K* in time *t* ($TProb_{p,k}^t$) are as follows:

$$Prob_{p,k}^t = sp(p,k,t) \times \Omega_{p,t}^t \times Intertia\,_k^t \times (1 - sc_{c \to k}) \tag{16}$$

$$\Omega_{p,t}^t = \frac{\sum\limits_{N \times N} con\left(c_p^{t-1} = k\right)}{N \times N - 1} \times \omega_k \tag{17}$$

where $sc_{c \to k}$ is conversion cost of land *c* to land *k*; $1 - sc_{c \to k}$ is the difficulty coefficient for land conversion; $\Omega_{p,t}^t$ is the neighborhood effect; $\sum\limits_{N \times N} con\left(c_p^{t-1} = k\right)$ represents the total number of land *k* at time $t-1$ under the mole window of $n \times n$ matrix, where *n* is 5 in this study; $\omega_k$ is the domain factor parameter of each class.

Finally, agricultural data of 2010 is used to simulate the data of 2015, and the simulation results are compared with the real situation to verify the accuracy of the model by the indexes of overall accuracy and *kappa* coefficient. Generally speaking, *kappa* < 0.5, the fitting effect is poor; $0.5 \leq kappa \leq 0.75$, the fitting effect is general; *kappa* > 0.75, the fitting effect is good.

## 3. Results and Analysis

### 3.1. Overall Characteristics of Agricultural Land Use Change

3.1.1. More Than Half of Total Extent Is Agricultural Land, While the Extent of Which Has Decreased

The extent of agriculture land in Xiangxi was 8454.98 km$^2$ in 2000 and 8323.72 km$^2$ in 2018 (Table 3), while the proportions of which, to the total extent in the above two years, were 58.40% and 54.66%, respectively. The total agricultural land extent, therefore, decreased about 131.26 km$^2$ or 3.74%. The above data showed that Xiangxi was dominated by agricultural production, but the problem of agricultural land occupation was a little serious.

**Table 3.** Agricultural land extent (km$^2$) between 2000 and 2018.

| Category | Arable Land | Forestry Land | Rangeland | Fishery Land | Total |
|---|---|---|---|---|---|
| 2000 | 2990.99 | 4193.75 | 1239.42 | 30.81 | 8454.98 |
| 2018 | 2925.80 | 4142.95 | 1232.24 | 22.72 | 8323.72 |
| Mean | 2958.40 | 4168.35 | 1235.83 | 26.77 | 8389.35 |
| net change, 2000–2018 | −65.19 | −50.80 | −7.18 | −8.09 | −131.26 |
| % of 2018 | 35.15 | 49.77 | 14.81 | 0.27 | 100 |

Forestry land was the most abundant class with an average extent of 4168.35 km$^2$ or a proportion of 49.77% in 2018, while fishery land was the rarest class with only 26.77 km$^2$ or 0.27%, reflecting that the study area had natural advantage in forestry production and the advantage for fishery production was insufficient. Each agricultural land had varied loss across these 18 years, and arable land had the highest loss of 65.19 km$^2$, while rangeland had the lowest of 7.18 km$^2$. A constant decrease of agricultural land undoubtedly threatened the food supply and brought the considerable economic pressure to farmers in poor areas [39], so related stakeholders should unswervingly protect the agricultural land. Therefore, the basic conditions and development orientation of agricultural production could be clearly reflected by the composition and change of agricultural land.

3.1.2. The Change of Arable Land and Fishery Land Was the Most Prominent

About 88.26% (7775.03 km$^2$) of agriculture land have never experienced land use change during the study period (Table 4, Figure 3), while the rest 11.47% (1033.94 km$^2$) experienced it at least once. Change was spatially heterogeneous. Arable land change totaled 47.93% of the total, and the change occurred across the region. Nearly 38.87% of the change involved forestry land, with the majority change occurring in the central region; 11.52% of the total change was comprised by rangeland, and this change occurred predominantly in southwest of Jishou, north of Yongshun, and Longshan. Fishery land use change comprised the rest at 1.68%, and was the most intensive in Huayuan and Jishou.

Arable land had the highest of change importance ($C_{impor}$) at 49.36%, while rangeland had the lowest of 5.64% (Table 4), reflecting the most obvious change and the least obvious change, respectively. At the same time, fishery land had the highest loss rate ($N_a$ was −26.27%), while rangeland was the lowest ($N_a$ was −0.6%). In addition, the index of dynamic ($K_d$) was also different, with the highest (0.77%) for arable land, while the lowest (0.09%) for rangeland. In short, the above data indicated that the change of arable land and fishery land was the most prominent.

**Table 4.** Agricultural land change between 2000 and 2018.

| Change of 2000–2018 | No Change | Arable Land Change | Forestry Land Change | Rangeland Change | Fishery Land Change |
|---|---|---|---|---|---|
| Area (km$^2$) | 7775.03 | 495.65 | 401.89 | 119.07 | 17.33 |
| % of agricultural land | 88.26 | 5.63 | 4.56 | 1.35 | 0.20 |
| % of change | | 47.93 | 38.87 | 11.52 | 1.68 |
| $C_{impor}$ (%) | | 49.36 | 38.88 | 5.64 | 6.12 |
| $N_a$ (%) | | −2.18 | −1.22 | −0.60 | −26.27 |
| $K_d$ (%) | | 0.77 | 0.61 | 0.09 | 0.10 |

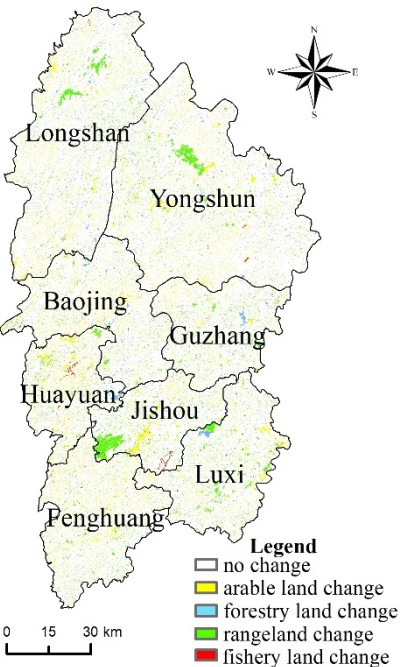

**Figure 3.** Spatial distribution of agricultural land change from 2000 to 2018.

*3.2. Characteristics of Each Agricultural Land Use Change*

3.2.1. The Density of Each Agricultural Land Had Strong Spatial Heterogeneity and Changed Slightly

The spatial patterns of arable land density in Xiangxi showed a trend of lower in the southeast and northwest, and higher in the center (Figure 4a). The higher-density areas concentrated in Yongshun, Baojing, Huayuan, and Fenghuang, while the lower-density areas were distributed in the junction of Jishou and Luxi. The reason is that the rivers flow through the central part and the terrain is flatter, so there is more arable land. In addition, the density dynamic of arable land had spatial heterogeneity, and Jishou has decreased faster than other regions. The reason was that Jishou was the political and economic center, and the problem of built-up area expansion at the expense of agricultural land was more prominent.

The spatial distribution of forestry land density in Xiangxi was higher in the southeast and northwest, and lower in the center (Figure 4b). The higher-value centers were Guzhang, Jishou, and Luxi, and the lower were Yongshun, Baojing, and Huayuan. It decreased in the north (Longshan and Yongshun) and increased in the south (Jishou), respectively. Overall, these results suggested that forestry land in the southwest and northwest was more lush and the environment was more beautiful, and this trend was strengthened in the south.

The density of rangeland in the northern and central regions was much higher than that of the southern region (Figure 4c). It presented a spatial pattern of four high-value centers surrounding a low-value center in the northern and central regions. At the same time, the rangeland density in the northeast of Yongshun and at the junction of Baojing and Huayuan was much higher than other areas. In addition, the spatial distribution of rangeland density in Xiangxi changed little from 2000 to 2018.

The density of fishery land in Xiangxi had a characteristic of higher in the south and lower in the north (Figure 4d). The high-value centers were mainly located in the northern area of Huayuan, the junction of Jishou and Luxi, and the eastern area of Luxi. During the study period, it showed a trend of stability in the north, and downward in the center and south. Specifically, it decreased most dramatically in the northern area of Huayuan and the junction of Jishou and Luxi; the bulk of this change was because of built-up area expansion which was more prevalent in the south.

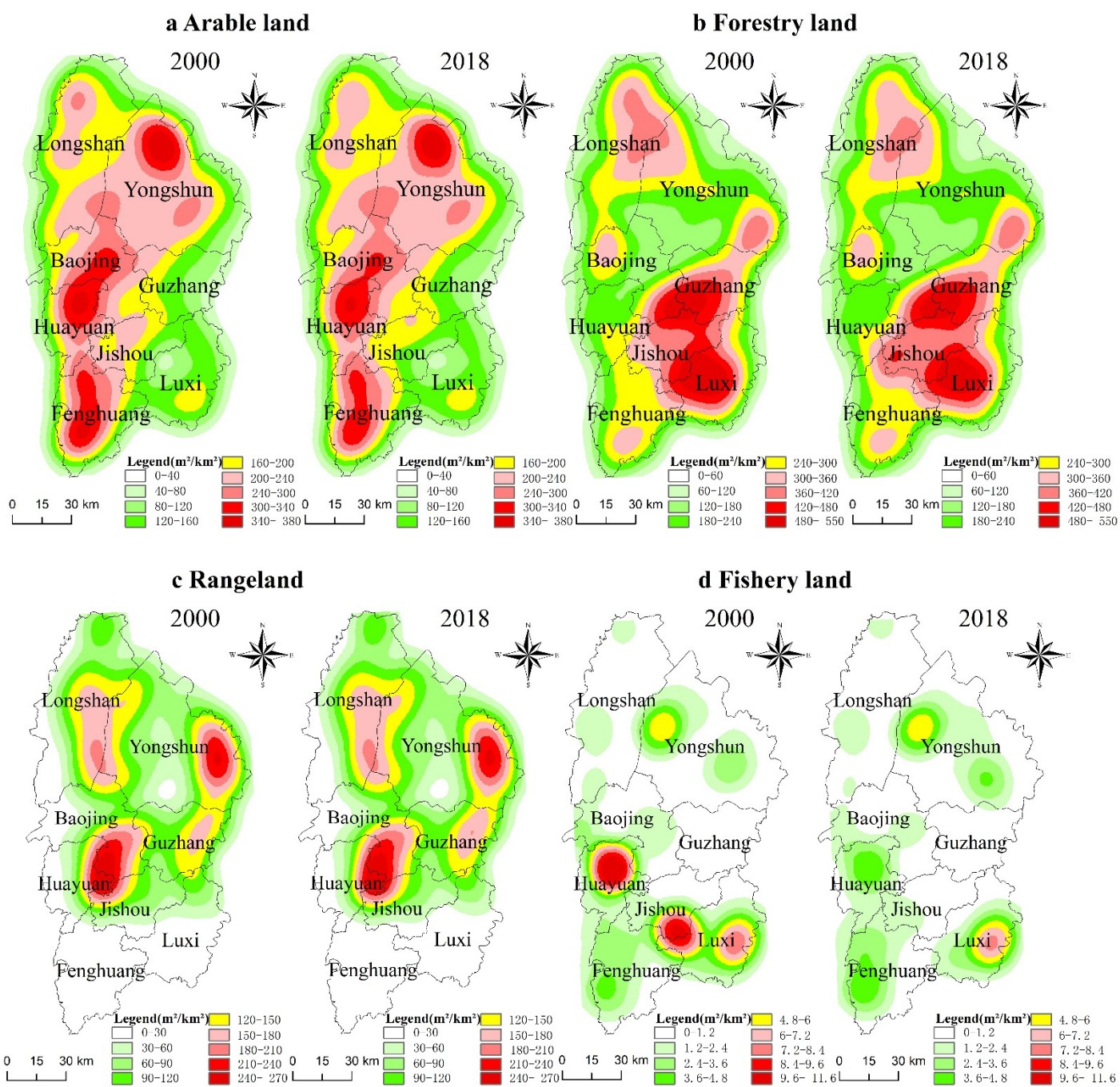

**Figure 4.** (**a**–**d**) The density of agricultural land in Xiangxi from 2000 to 2018.

### 3.2.2. Main Change Trend Was Loss, and Change Had Spatial Heterogeneity

The gain of arable land was concentrated in the north, and Yongshun had the highest positive gain (62.95 km$^2$) (Figure 5a, Table 5). Nevertheless, the loss was concentrated in the south and north, and Yongshun also had the highest negative loss (75.17 km$^2$) (Figure 5a, Table 5). Arable land in all regions declined across the 18 years, and Jishou declined the most (20.55 km$^2$), while Huayuan declined the least (2.90 km$^2$). Yongshun and Jishou were the most frequently changing regions (consisting of 47.42% of total change), while Huayuan and Longshan were the least frequently changing regions (only consisting of 8.8%) (Table 5). In addition, the change rate ($N_a$) and dynamic ($K_d$) in Jishou were also higher than other areas. This showed that the problem of arable land occupation in Xiangxi could not be ignored, especially in the administrative center (Jishou).

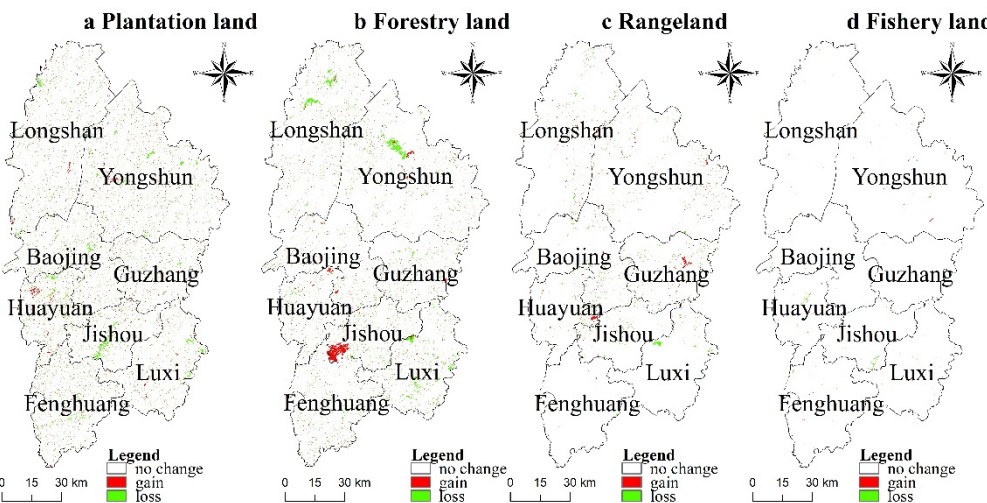

**Figure 5.** (**a–d**) Spatial distribution of loss and gain for each agricultural land for 2000–2018.

The gain or loss of forestry land had spatial differences (Figure 5b, Table 5). The gain was concentrated in the south, and Jishou had the highest positive gain (60.70 km$^2$), while Huayuan had the lowest (16.54 km$^2$). The loss was more obvious in the north, and the highest negative loss was in Yongshun (66.33 km$^2$), while Huayuan the lowest (16.38 km$^2$). Forestry land in majority areas reduced, except for Huayuan and Jishou. Nearly 95% of forestry land change occurred in Jishou, Luxi, Longshan, and Yongshun ($C_{impor}$ were 30.17%, 20.46%, 20.03%, and 23.01%, respectively). $N_a$ in Jishou was 10.27%, which was the fastest growth rate, while Luxi had $-4.54$%, the most significant shrinkage. $K_d$ in Jishou, Luxi, Longshan, and Yongshun were also higher than other regions. Above all, forestry land in most administrative units decreased, and the magnitude of the decline varied, which was higher in north and south, and lower in the center.

The loss of rangeland was concentrated in the south, and the gain in the center (Figure 5c). The gain and loss in Yongshun were 24.14 km$^2$ and 22.28 km$^2$, which were highest gain and highest loss (Table 5), respectively. Since 2000, rangeland in most regions decreased, except for Yongshun, Huayuan, and Guzhang. Furthermore, Guzhang and Luxi exhibited larger change, with the highest gain in Guzhang (5.20 km$^2$) and the highest loss in Luxi ($-8.32$ km$^2$). During the 18-year period, Guzhang and Luxi were the most frequently changing areas, consisting of nearly 50% of total change, while Fenghuang was the least, with only 1.65%. Guzhang had the fastest growth ($N_a$ is 3.11%), while Luxi had fastest shrinkage ($N_a$ is $-16.79$%). In addition, $K_d$ in Luxi and Guzhang were also much higher than other regions. In short, the main trend of rangeland in Xiangxi was shrinkage, and Luxi and Guzhang were the typical case of growth and shrinkage, respectively.

The gain and loss of fishery land varied across regions (Figure 5d). The gained areas were sparsely distributed, Yongshun had the highest positive gain (2.23 km$^2$), and Guzhang the lowest (0.08 km$^2$) (Figure 5a, Table 5). Nevertheless, the lost areas were concentrated in the south, Huayuan had the highest negative loss (3.90 km$^2$), and Guzhang the lowest (0.45 km$^2$). Since 2000, fishery land in most regions had shrunk, with the exception of Yongshun. Jishou and Huayuan had the most significant shrinkage (3.01 km$^2$ and 3.43 km$^2$, respectively). Likewise, the most frequent change of fishery land occurred in Huayuan and Jishou, consisting of nearly 65% of total change. As for $N_a$, most regions decreased by more than 10%, and even more than 50% in some regions, indicating there were serious problems of ponds and reservoirs' occupation. $K_d$ in Huayuan and Jishou were also much higher than other regions. Overall, fishery land in Xiangxi had declined since 2000, with the most obvious decline in Jishou and Huayuan.

**Table 5.** Statistics of each agricultural land use change for 2000–2018.

| | | | Arable Land | | | |
|---|---|---|---|---|---|---|
| **Name** | **Gain (km$^2$)** | **Loss (km$^2$)** | **Net Change (km$^2$)** | $C_{impor}$ **(%)** | $N_a$ **(%)** | $K_d$ **(%)** |
| Yongshun | 62.95 | 75.17 | −12.22 | 17.53 | −1.43 | 0.38 |
| Baojing | 33.22 | 40.51 | −7.29 | 10.76 | −1.98 | 0.23 |
| Huayuan | 34.28 | 37.18 | −2.90 | 4.40 | −1.06 | 0.10 |
| Jishou | 17.37 | 37.91 | −20.54 | 29.89 | −11.96 | 0.65 |
| Guzhang | 20.44 | 26.40 | −5.96 | 8.59 | −3.31 | 0.19 |
| Luxi | 23.61 | 31.40 | −7.79 | 11.21 | −3.14 | 0.24 |
| Longshan | 46.06 | 49.03 | −2.97 | 4.40 | −0.49 | 0.10 |
| Fenghuang | 31.09 | 40.09 | −9.01 | 13.23 | −2.05 | 0.29 |
| | | | Forestry land | | | |
| **Name** | **Gain (km$^2$)** | **Loss (km$^2$)** | **Net change (km$^2$)** | $C_{impor}$ **(%)** | $N_a$ **(%)** | $K_d$ **(%)** |
| Yongshun | 34.92 | 66.33 | −31.41 | 23.01 | −3.51 | 0.71 |
| Baojing | 19.45 | 19.73 | −0.28 | 0.19 | −0.06 | 0.01 |
| Huayuan | 16.54 | 16.38 | 0.16 | 0.02 | 0.01 | 0.00 |
| Jishou | 60.70 | 19.52 | 41.18 | 30.17 | 10.27 | 0.93 |
| Guzhang | 17.22 | 23.18 | −5.96 | 4.33 | −1.26 | 0.13 |
| Luxi | 20.40 | 48.38 | −27.98 | 20.46 | −4.54 | 0.63 |
| Longshan | 27.72 | 54.90 | −27.18 | 20.03 | −3.35 | 0.62 |
| Fenghuang | 24.13 | 26.58 | −2.45 | 1.78 | −0.47 | 0.05 |
| | | | Rangeland | | | |
| **Name** | **Gain (km$^2$)** | **Loss (km$^2$)** | **Net change (km$^2$)** | $C_{impor}$ **(%)** | $N_a$ **(%)** | $K_d$ **(%)** |
| Yongshun | 24.14 | 22.28 | 1.87 | 6.24 | 0.39 | 0.13 |
| Baojing | 9.39 | 12.75 | −3.36 | 12.99 | −2.19 | 0.27 |
| Huayuan | 10.70 | 8.07 | 2.63 | 9.76 | 2.33 | 0.20 |
| Jishou | 5.33 | 7.77 | −2.44 | 9.34 | −3.01 | 0.19 |
| Guzhang | 16.57 | 11.37 | 5.20 | 19.40 | 3.11 | 0.40 |
| Luxi | 2.23 | 10.55 | −8.32 | 30.79 | −16.79 | 0.64 |
| Longshan | 12.27 | 14.89 | −2.62 | 9.83 | −0.93 | 0.20 |
| Fenghuang | 0.42 | 0.86 | −0.44 | 1.65 | −6.93 | 0.03 |
| | | | Fishery land | | | |
| **Name** | **Gain (km$^2$)** | **Loss (km$^2$)** | **Net change (km$^2$)** | $C_{impor}$ **(%)** | $N_a$ **(%)** | $K_d$ **(%)** |
| Yongshun | 2.23 | 2.08 | 0.15 | 1.50 | 2.19 | 0.41 |
| Baojing | 0.51 | 1.25 | −0.74 | 7.97 | −28.04 | 2.16 |
| Huayuan | 0.86 | 3.90 | −3.04 | 34.35 | −53.55 | 9.30 |
| Jishou | 0.29 | 3.66 | −3.38 | 39.11 | −72.97 | 10.59 |
| Guzhang | 0.08 | 0.45 | −0.37 | 4.02 | −55.60 | 1.09 |
| Luxi | 0.76 | 1.63 | −0.88 | 10.02 | −13.85 | 2.71 |
| Longshan | 0.46 | 0.74 | −0.28 | 2.96 | −10.80 | 0.80 |
| Fenghuang | 1.30 | 1.31 | −0.01 | 0.07 | −0.15 | 0.02 |

*3.3. The Future Scenario of Agricultural Land Use in Xiangxi*

The simulation accuracy of arable land, forestry land, rangeland, and fishery land are 95.95%, 96.54%, 96.85%, and 85.33%, respectively. At the same time, the overall accuracy and the *Kappa* are 96.33% and 0.92, respectively. The above data show that the accuracy has reached a high level, and the FLUS model has strong applicability in this study.

Agricultural land of Xiangxi in 2030 will maintain the existing spatial attributes (Figure 6): arable land shows a northeast-southwest pattern; forestry land is mainly concentrated in the southeast; rangeland is more concentrated in north than the south; fishery land is scattered in the whole area, and more in the south. That is roughly the same as 2000 and 2018.

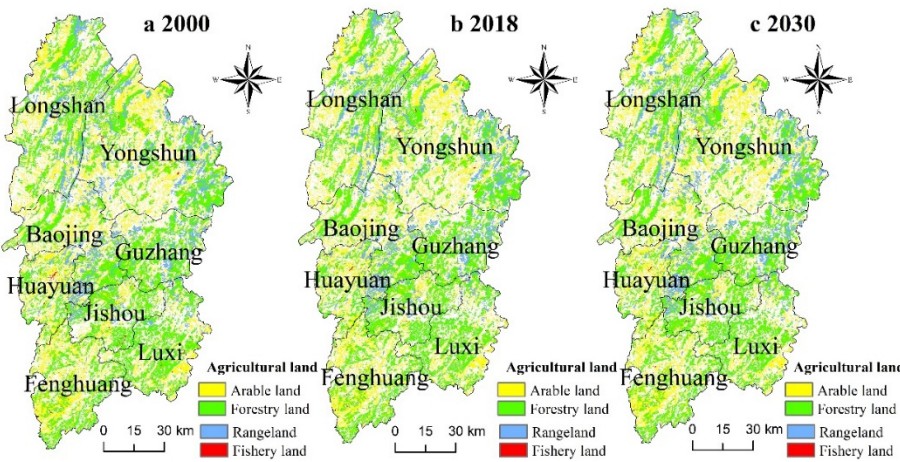

**Figure 6.** (**a–c**) Spatial pattern of agricultural land use in Xiangxi from 2000 to 2030.

All types of agricultural land will decrease in 2030, and the extent of arable land has the highest loss of 113.47 km$^2$ as opposed to 2000, while fishery land has the lowest of 15.34 km$^2$ (Table 6). However, fishery land will have the highest negative loss rate (−49.78%), and rangeland the lowest negative loss rate (−1.86%). The above data shows that agricultural land change is mainly composed of arable land and fishery land. The reason is that the area where arable land and fishery land are located is more flat, with a larger number of population, which is more susceptible to interference by human activities. However, the composition of agricultural land was almost unchanged: forestry land had the largest proportion (49.97%), followed by arable land and rangeland (35.03% and 14.81%, respectively), and fishery land the smallest (0.19%). Overall, both of the area and proportion of agricultural land will decline in 2030, with a loss of 241.34 km$^2$ or 2.85% decrease from 2000.

**Table 6.** Statistics of agricultural land in Xiangxi in 2030.

| Agricultural Land | Arable Land | Forestry Land | Rangeland | Fishery Land | Total |
|---|---|---|---|---|---|
| area (km$^2$) | 2877.52 | 4104.28 | 1216.37 | 15.47 | 8213.65 |
| proportion (%) | 35.03 | 49.97 | 14.81 | 0.19 | 100 |
| net change (km$^2$) | −113.47 | −89.47 | −23.05 | −15.34 | −241.34 |
| change rate (%) | −3.79 | −2.13 | −1.86 | −49.78 | −2.85 |

Note: Both net change and change rate were obtained by comparing with 2000.

In summary, agricultural land in Xiangxi will be slightly occupied in 2030, but there are certain differences in the amount and intensity of each agricultural land. Social and economic development have led to higher demand for built-up more than agricultural land, resulting in shrinkage of agricultural land, and expansion in construction land. However, the Chinese government has issued strict protection policies of agricultural land, which curb the disorderly construction expansion to a certain extent. Therefore, the slight shrinkage of agricultural land in Xiangxi is in line with the objective law, under the current socio-economic background and policy regulations.

## 4. Conclusions and Discussion

### 4.1. Conclusions

It was found that Xiangxi was dominated by agricultural land, but the total extent of which decreased by about 56.89 km$^2$ or 3.74% from 2000 to 2018. In terms of the composition, nearly 50% of total extent was forestry land, reflecting great potential advantage of forestry production in the study area. Since 2000, the area of each agricultural land reduced, with the most obvious reduction in arable land.

The density of arable land in the study area showed a trend of lower in the southeast and northwest, and higher in the middle, with shrinkage in most units, especially in the administrative center. The density spatial pattern of forestry land was just the opposite of

arable land and decreased in most regions, with more dramatic change in northwestern and southwestern areas. The density of rangeland in the northern and central regions was much higher than that of the southern regions; both trends of decrease and increase coexisted, with the main trend being decrease. The density of fishery land was higher in the north and lower in the south, with a trend of stability in the north and decrease in the center and south.

In 2030, agricultural land in Xiangxi will almost maintain the existing spatial pattern, and the extents of the arable land and fishery land have the highest loss of 113.47 km$^2$ and the lowest loss of 15.34 km$^2$ as opposed to 2000, respectively. The composition of agricultural land will become nearly static, and both the area and proportion will decline in 2030, with a loss of 241.34 km$^2$ or 2.85% decrease from 2000. However, the slight shrinkage of the agricultural land in Xiangxi is in line with the objective law.

### 4.2. Discussion

4.2.1. Implications for Agricultural Land Use

In this study, we aimed to analyze the dynamic pattern of agricultural land use and predict its future scenario in Xiangxi. Xiangxi is an important region of characteristic agricultural products, with a large area and a high proportion of agricultural land. At the same time, Xiangxi is also significantly affected by the policies of new urbanization, beautiful village construction, and village revitalization. In terms of the analysis results, agricultural land in Xiangxi has been occupied, with the most prevalent occupation in Jishou (the administrative center). What is more, the problem of its occupation is consistent with that in eastern China [40,41] and Hunan Province [42], but not as obvious as that in Changsha [43].

As far as the simulation results, the distribution pattern and change characteristics of the agricultural land in Xiangxi are generally in line with the requirements of socio-economic development and agricultural land protection policies. However, the problem of agricultural land occupation will also exist in the future, which is similar to some other regions of the world [44–46]. In addition, the occupation of arable land and fishery land is more serious.

Therefore, it is necessary to strengthen the scientific management and rational utilization of agricultural land, with emphasis on arable land and fishery land in the south, especially the administrative center, to minimize the impact of the construction land expansion [47,48].

4.2.2. Contribution of This Study

Compared with existing studies [1,2], a scientific, systematic, and reasonable methodology has been proposed. First, based on the existing research and land use data, the agricultural land classification system is constructed. Second, GIS spatial analysis methods, nuclear density analysis, and the indexes of change importance, change area, change rate, and dynamic are used to explore the distribution pattern and the evolution characteristics of agricultural land in Xiangxi from 2000 to 2018. It realizes the long time and fine evaluation of agricultural land use change, and can provide support for rural land planning and reference for research on similar topics.

Gray prediction model and GeoSoS-FLUS model were used to simulate agricultural land in Xiangxi in 2030. The simulation process has not only considered agricultural land change since 2000, but also taken into account the key driving factors. The predicted results are very similar to the actual agricultural land use, indicating that the combination of the two models has good applicability and feasibility. This research has enriched the theory and methods of land use change and its simulation.

4.2.3. Research Prospect

Nevertheless, we have not taken into account the adjustment of development strategies and the changes of their focus and direction, and we did not carry out the simulation under multiple scenarios. Therefore, in the future, we will adjust and modify the parameters,

and explore agricultural land use under different conditions to find the best solution of scientific planning and rational management for agricultural land.

**Author Contributions:** Q.Y. designed this study. H.X., Y.M., R.Z. and H.C. collected and analyzed the data. H.X. wrote the manuscript. Q.Y. reviewed the manuscript. All authors have read and agreed to the published version of the manuscript.

**Funding:** This research was funded by the Key Project of Chongqing Key Research Base of Humanities and Social Sciences, grant number 14SKB014.

**Data Availability Statement:** The datasets used and/or analyzed during the current study are available from the corresponding author on reasonable request.

**Acknowledgments:** We thank the employees who work in agricultural sector in Xiangxi for their support and valuable data. We also acknowledge the financial support from the research group.

**Conflicts of Interest:** The authors of this article declare no potential conflict of interest.

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
