# Peer review of "Spatio-Temporal Evolution and Future Simulation of Agricultural Land Use in Xiangxi, Central China"

_land, doi:10.3390/land11040587_

Round 1
Reviewer 1 Report
The paper quantify the land use change in central China and based on the available trends future projections have been made
Please make change in the article as per the attached pdf file
The approach used for quantifying the change and rate of change in specifically homogenous land use in adequate for course of study. However authors may consider to include the impact of development strategy on those changes subsequently indicated in their conclusion

Author Response
Dear Reviewer:
Thanks for your letter and the comments concerning our manuscript. Those comments are all valuable and very helpful for revising and improving our paper, as well as the important guiding significance to our researches. We have studied comments carefully and have made correction which we hope meet with approval. Revised contents are marked using the “Track Changes” function of Microsoft Word in our revised manuscript. The point to point responses to your comments are listed as following (revised contents are marked as red). Thank you for your review again.

Reviewer 2 Report
Congratulations for the work.
I suggest to enlarge a little the discussion section. Comparing it with the results chapter it is too short.
I suggest enlarging the size of the captions of the images because it is not read well, otherwise I have no other comments.
Author Response
Dear Reviewer:
Thank you very much for reviewing our manuscript and for giving us an opportunity to revise it. We really appreciate your positive and constructive comments and suggestions, which are very helpful to improve our manuscript. After serious consideration, we have adopted all suggestions and made according changes. Following the revisions, we believe that the quality of the manuscript has been improved considerably, and we hope that it can meet the publication standard of land.
Revised contents are marked using the “Track Changes” function of Microsoft Word in our revised manuscript. The point to point responses to your comments are listed as following (revised contents are marked as red). Thank you for your review again.

Reviewer 3 Report
Based on the spatio-temporal change of agricultural land, this paper explores the pattern and characteristics of each type of agricultural land and administrative unit from 2000 to 2018. The simulation of agricultural land use in 2030 is obtained through Grey Forecasting and GeoSoS-FLUS model, and the corresponding features of agricultural land use are then evaluated. Relevant suggestions are posed to the land use, which is crucial to human activities and social development. The specific comments are as follows,
Introduction
There are too many descriptions about the definition and importance of the agricultural land, while the literatures review is not focused, and the contribution to the literatures is not clear.
Data and method
The process of 4 classifications of agricultural land use is vague, especially for the basis for classification in the table 1. Is it the basis of the classification by the Land and Resources Ministry of China, or the procedure of this studies? If this paper doesn’t do any changes about the original data, then the basis seems to redundant due to the Resources and Environmental Sciences Data Center of the Chinese Academy of Sciences has done the job.
Is the unit of the driving factor consistent with the land use data, as the population data is from the statistic yearbook?
Results
The content and headline in 3.1(2) and 3.2 seem to be overlapping, and the analysis is a little wordy. The indicators in the table 4, % of change and Cimpor have no significant differences. The content in 3.2(2) is too detailed and no important meanings and findings can be seen. The structure can be reorganized scientifically.
In addition, some statements should be corrected, such as:
“lagging economic” is grammarly wrong in the sentence “As a traditional area with lagging economic,”
“Yongshu” is wrongly spelled in “Nevertheless, the loss was concentrated in the south and North (such as 323 Jishou and Yongshu)” in 3.2;
And, in 3.2(1) the cultivated land and cropland, forestry land and woodland are advised to be unified.
Discussion
“However, the problem of agricultural land occupation will also exist in the future, which is similar to the predicted results in some parts of the world[38-40].” This sentence seems to be logically confused, and the references here have no relevance?
Author Response
Dear Reviewer:
Thank you very much for reviewing our manuscript and for giving us an opportunity to revise it. We really appreciate your positive and constructive comments and suggestions, which are very helpful. After serious consideration, we have adopted all suggestions and made according changes. Following the revisions, we believe that the quality of the manuscript has been improved considerably, and we hope that it can meet the publication standard.
Revised contents are marked using the “Track Changes” function of Microsoft Word in our revised manuscript. The point to point responses to your comments are listed as following (revised contents are marked as red). Thank you for your review again.

Round 2
Reviewer 3 Report
All comments have been revised and responded by the authors , and I think the article could be accepted now.